# Small tandem DNA duplications result from CST-guided Pol α-primase action at DNA break termini

Joost Schimmel[1], Núria Muñoz-Subirana[1], Hanneke Kool[1], Robin van Schendel [1] & Marcel Tijsterman [1,2✉]

Small tandem duplications of DNA occur frequently in the human genome and are implicated in the aetiology of certain human cancers. Recent studies have suggested that DNA double-strand breaks are causal to this mutational class, but the underlying mechanism remains elusive. Here, we identify a crucial role for DNA polymerase α (Pol α)-primase in tandem duplication formation at breaks having complementary 3′ ssDNA protrusions. By including so-called primase deserts in CRISPR/Cas9-induced DNA break configurations, we reveal that fill-in synthesis preferentially starts at the 3′ tip, and find this activity to be dependent on 53BP1, and the CTC1-STN1-TEN1 (CST) and Shieldin complexes. This axis generates near-blunt ends specifically at DNA breaks with 3′ overhangs, which are subsequently repaired by non-homologous end-joining. Our study provides a mechanistic explanation for a mutational signature abundantly observed in the genomes of species and cancer cells.

[1] Department of Human Genetics, Leiden University Medical Center, Leiden, The Netherlands. [2] Institute of Biology Leiden, Leiden University, Leiden, The Netherlands. ✉email: m.tijsterman@lumc.nl

Small Tandem Duplications (TDs) of DNA are the most common form of short (<100 bp) insertions occurring in the human genome[1], thereby contributing to genome expansion and evolution but also resulting in novel gene functions potentially causing human diseases. FLT3-internal tandem duplications, for example, are driving acute myeloid leukemia (AML) and their occurrence is linked to a poor prognosis[2]. Although most profound for FLT3, the role of TDs in the etiology of cancer is not limited to this gene: similar activating mutations have been observed in other oncogenes[3–8]. In addition, the prevalence of this mutation type is likely being underestimated as small TDs are often discarded by mapping algorithms used to analyze next-generation sequencing data[4]. In contrast to an increased understanding of the molecular mechanisms underlying larger segmental duplications (1–100 kb)[9–12], we still have very little knowledge concerning the etiology of small-sized duplications. Early work suggested unequal sister chromatid exchange and polymerase slippage during replication to account for these types of mutations[13]. However, bioinformatic analysis of naturally occurring DNA insertions in humans and plants has hinted towards a model where TDs arise through error-prone repair of DNA double-strand breaks (DSBs) that may result from nearby single-strand breaks in opposing strands[1,14]. This model was recently supported by the observation that TDs abundantly form at DSBs with small overhangs induced by Cas9-nickase enzymes in different species[15–17]. Importantly, we found that TDs preferentially form at DSBs with 3′ protruding ends and require Ku80, a core component of the non-homologous end-joining pathway (NHEJ)[16]. Based on these findings, we hypothesized that TD formation might be the result of fill-in synthesis taking place on both complementary overhangs prior to end-joining. Which enzymes underlie such potential fill-in synthesis has remained elusive since TDs were formed at a comparable frequency upon knockouts of the NHEJ DNA polymerases Lambda and Mu, or the alt-EJ DNA polymerase Theta.

At telomeres, replication-induced long 3′ ssDNA ends are filled in by the CST-Pol α-primase complex to avoid DNA loss[18]. Intriguingly, this complex has also been linked to canonical NHEJ as it is recruited to sites of DNA damage by the chromatin-binding protein 53BP1 and the recently identified Shieldin complex[19–24]. This work proposed that CST-Polα together with Shieldin components protects the break ends from extensive resection.

Here, we demonstrate that CST-Polα can initiate close to the termini of DSBs with 3′ protruding ends in a 53BP1 and Shieldin dependent manner. We show that this activity induces erroneous duplication of genomic sequences at DSBs with complementary ends, resulting in small DNA duplications similar to those observed in human pathologies.

## Results

### CST-complex promotes tandem duplication formation at breaks with complementary ends

Previously, we have established an assay that reads out mutagenic repair of CRISPR/Cas9-induced breaks by exploiting the X-linked selectable marker gene HPRT[16]. To improve the accuracy and versatility of this assay, we placed the eGFP-coding sequence downstream of the endogenous HPRT locus (in the frame to its last exon), resulting in a functional eGFP tagged HPRT protein in mouse embryonic stem (mES) cells (Fig. 1a, Supplementary Fig. 1a). These mES cells now allow us to monitor the mutation frequency of Cas9-induced breaks at any site within the HPRT-eGFP open reading frame, by quantifying the number of GFP-negative cells in the population (Fig. 1b, Supplementary Fig. 1b, see "Methods" section). To test the potential involvement of the CST-complex in the formation of

tandem duplications we previously observed at CRISPR/Cas9-induced breaks (Fig. 1c,[16]), we generated CST-deficient mES cells by knocking out either Ctc1 (Supplementary Data 1) or Stn1 (Supplementary Data 1, Supplementary Fig. 2a). To monitor the end-joining of DSBs with 3′ ssDNA protrusions, we used Cas9-N863A to introduce nicks in opposite strands of eGFP, resulting in DSBs with 43 bp overhangs (Fig. 1a, Supplementary Fig. 1b)[25]. Quantification of GFP-negative cells after expression of sgRNAs and Cas9-N863A confirmed that mutagenic repair of introduced DSBs depends on Ku80, a core component of NHEJ (Fig. 1d)[16,26]. Both Ctc1, as well as Stn1 knockout clones, show a very similar reduction in mutation frequency, suggesting that the CST complex is required to repair DSBs with 3′ overhangs presumably due to its role in NHEJ (Fig. 1d, Supplementary Fig. 2c)[21,24]. Previous work identified DNA polymerase Theta (Pol Θ, encoded by the Polq gene)[16,27–29] to be responsible for mutagenic repair in NHEJ deficient cells, and deficiency of polymerase Theta-Mediated End-Joining (TMEJ) resulted in synthetic sickness when co-depleted with NHEJ[30]. We generated Polq-Ctc1 double knockout mES cells and found mutagenic repair to be largely abolished (Fig. 1d). At this stage, the fate of these Cas9-induced DSBs in cells deficient for CTC1 and TMEJ is unknown, but could potentially be error-free repair (e.g. by reannealing of the complementary protruding ends), or non-repair, triggering cell death.

To study the effect of the CST-complex on the repair-outcome of Cas9-N863A-induced breaks, we performed targeted sequencing on HPRT-eGFP mutant cells and compared the mutational profiles obtained from Pol Θ, Ku80 and CTC1 deficient cells to those from wild-type cells (Fig. 1e–g). Wild-type and Polq⁻/⁻ cells have very similar repair patterns: the majority of mutagenic repair in those cells represents tandem duplications (60.9% and 57.0%, respectively) pointing towards fill-in synthesis of seemingly stable 3′ ssDNA molecules. The strong reduction of tandem duplications in Ku80 deficient cells (11.7%) demonstrates NHEJ requirement, which in wild-type cells thus acts dominantly over TMEJ at these substrates. We observed a ~10-fold reduction in the number of tandem duplications in Ctc1⁻/⁻ cells as compared to wild-type cells, implicating the CST-complex in mutagenic processing of these DSB configurations (Fig. 1e, g). Tandem duplications were also greatly reduced in STN1 deficient cells (Supplementary Fig. 2d, f). In the absence of Ku80, CTC1, and STN1, residual mutagenic break repair is associated with more substantial loss of DNA flanking the DNA break, resulting in deletions that are characterized by microhomology at the junctions—a hallmark of TMEJ (Fig. 1f, g, Supplementary Fig. 2e, f). To address potential influences of sequence context we next analyzed repair of DSBs with 50 bp 3′ overhangs in HPRT exon 3 (Supplementary Fig. 3). This substrate produced outcomes highly similar to what we observed at the eGFP site (Supplementary Fig. 3a), although here the capacity of TMEJ to form tandem duplications in the absence of Ku80, CTC1 or STN1 is higher (Supplementary Fig. 3b). Notably, TDs generated by TMEJ have different characteristics than the majority of TDs appearing in wild-type cells, being larger in size (encompassing almost the entire 3′ protruding sequence) and much more frequently displaying microhomology at the junction (Supplementary Fig. 3c–f). We conclude that in wild-type cells, small tandem duplications arising at DSBs with 3′ complementary ends predominantly result from CST-dependent NHEJ; in its absence, TMEJ can partly compensate, guided by available microhomologies in the overhangs[30,31].

### The contribution of CST to NHEJ is DNA break-configuration specific

The CST complex regulates telomere maintenance through the recruitment of Pol α-primase to replicated telomeres

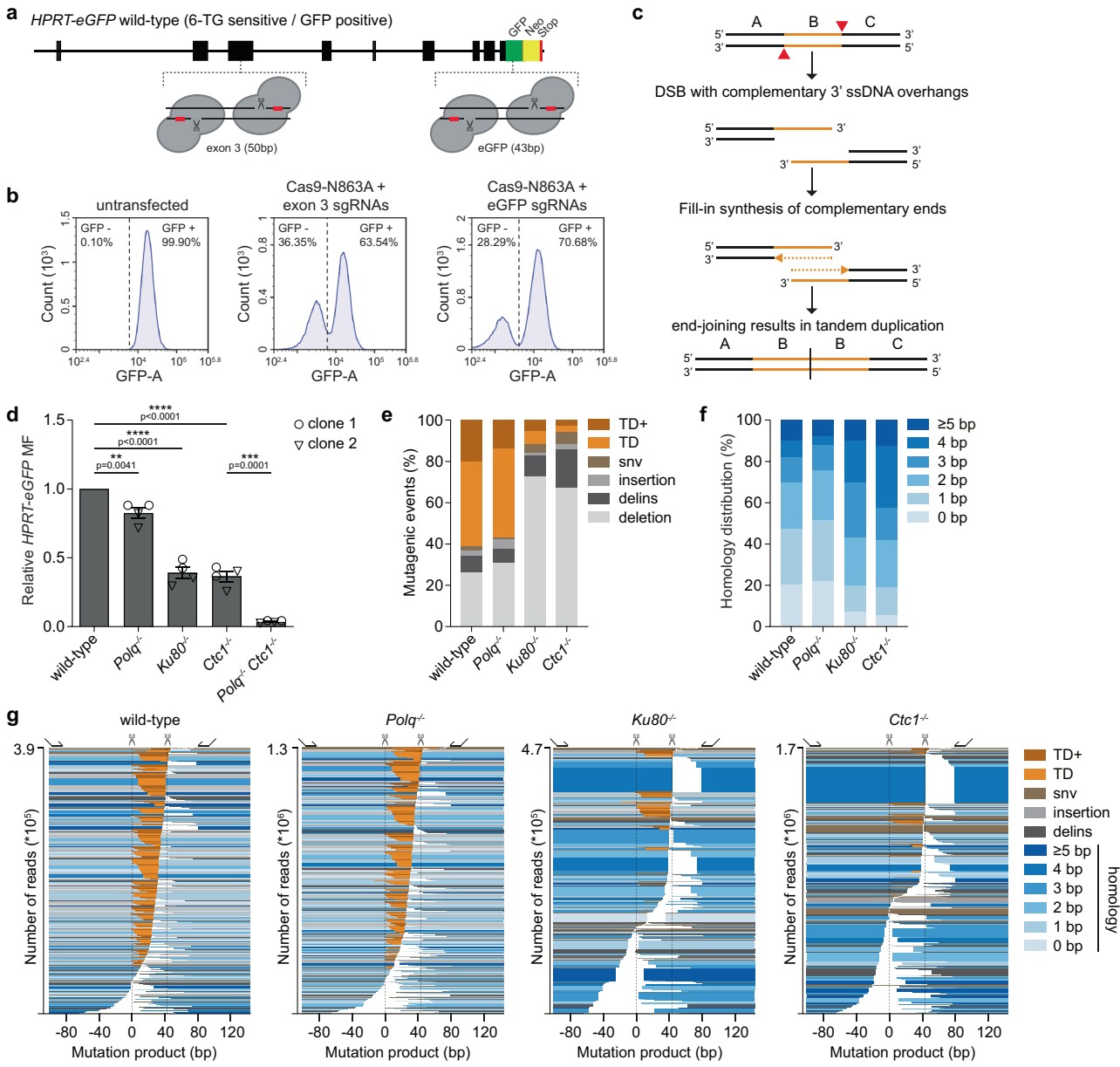

**Fig. 1 CST-complex promotes tandem duplication formation at DSBs with 3′ ssDNA overhangs. a** Schematic representation of the generated *HPRT-eGFP* gene and the two main target sites used in this study (exon 3 and eGFP, see Supplementary Fig. 1 for details). **b** FACS-plots showing the number of GFP-positive and GFP-negative cells in untransfected wild-type HPRT-eGFP cells and in HPRT-eGFP cells transfected with Cas9-N863A and two sgRNAs targeting exon 3 or *eGFP* seven days post-transfection. **c** Graphic illustration for tandem duplication formation at DNA-breaks with complementary 3′ ssDNA overhangs. **d** Relative *HPRT-eGFP* mutation frequency for the indicated mES cell lines (using two independent clones per genotype) transfected with Cas9-N863A and sgRNAs targeting *eGFP*, producing DSBs with 3′ ssDNA overhangs. The data shown represent the mean ± SEM ($n = 4$) and are expressed as a fraction of the mutation frequency observed in wild-type cells (set to 1). Statistical significance was calculated via a two-tailed unpaired *t*-test. **$P \leq 0.01$, ***$P \leq 0.001$ ****$P \leq 0.0001$. **e** Quantification of the types of mutagenic events (see "Methods" section for details) at breaks with 3′ ssDNA overhangs for the indicated mES cell-lines transfected with Cas9-N863A and sgRNAs targeting *eGFP*. GFP-negative (mutant) cells were sorted and used for targeted sequencing around the break-site, data represent the average of two independent clones per genotype. **f** Quantification of the extent of microhomology used for deletions and tandem duplications (TD) in the indicated genotypes. **g** Mutational signatures from (**e**) plotted as bars, relative to the Cas9-N863A-induced nicks (dashed lines), sorted by start position. The degree of blue coloring reflects the extent of microhomology. For TDs the duplicated sequence is plotted in orange. The height of each bar reflects the contribution of each outcome in the total amount of reads (*Y*-axis). DSB, double-strand break; MF, mutation frequency; delins, deletion insertion; snv, single nucleotide variant; TD, tandem duplication; TD+, tandem duplication with additional mutation (see "Methods" section); bp, base pair.

with long 3′ ssDNA overhangs[18,32–34]. Here, CST preferentially binds to ss-dsDNA junctions with 3′ protruding ssDNA generated by the 5′ to 3′ exonuclease activity of Exo1. In vitro, however, CST binds substrates containing 3′ or 5′ overhangs with similar efficiency[35]. To assess whether the role for the CST-complex in NHEJ is DSB-configuration selective, i.e., specifically required to process DSBs with 3′ ssDNA protrusions, we performed experiments using Cas9-D10A to make DSBs with 5′ overhangs in *HPRT* exon 3. We found that TMEJ and NHEJ act largely redundant on these types of breaks (Fig. 2a)[16]. The mutational

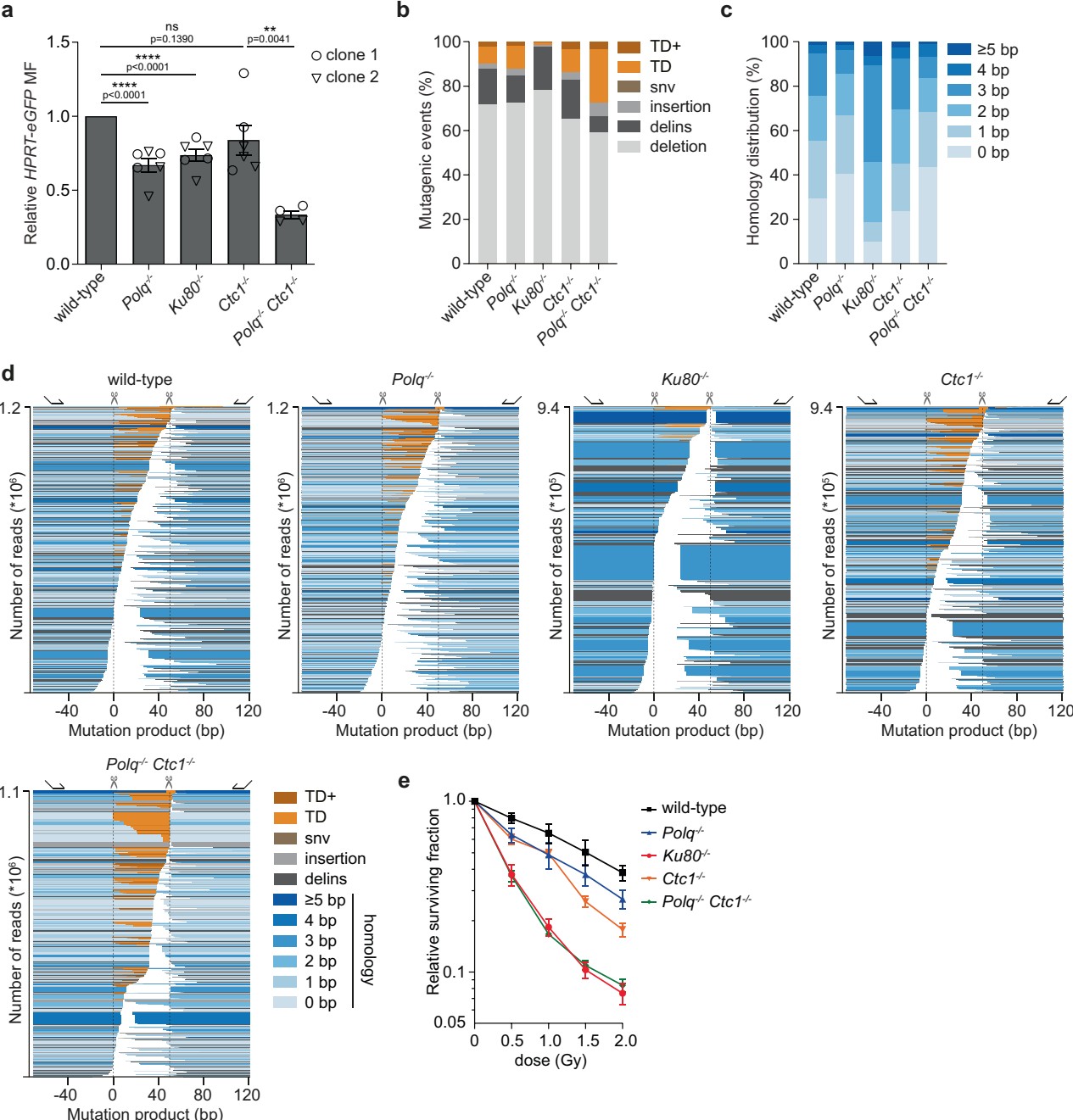

**Fig. 2 Contribution of CST to NHEJ is DNA break configuration specific. a** Relative *HPRT-eGFP* mutation frequency for the indicated mES cell-lines (using two independent clones per genotype) transfected with Cas9-D10A and sgRNAs targeting *HPRT* exon 3, producing DSBs with 5′ ssDNA overhangs. The data shown represent the mean ± SEM (*n* = 6, *n* = 4 for *Polq*−/− *Ctc1*−/− cells) and are expressed as a fraction of the mutation frequency observed in wild-type cells (set to 1). Statistical significance was calculated via a two-tailed unpaired t-test. ns, not significant (*P* > 0.05), \*\**P* ≤ 0.01, \*\*\*\**P* ≤ 0.0001. **b** Quantification of the types of mutagenic events at breaks with 5′ ssDNA overhangs for the indicated mES cell lines transfected with Cas9-D10A and sgRNAs targeting *HPRT* exon 3. GFP-negative (mutant) cells were sorted and used for targeted sequencing around the break-site, data represent the average of two independent clones per genotype. **c** Quantification of the extent of microhomology used for deletions and tandem duplications (TD) in the indicated genotypes. **d** Mutational signatures from (**b**) plotted as bars, relative to the Cas9-D10A-induced nicks (dashed lines), sorted by start position. The degree of blue coloring reflects the extent of microhomology. For TDs the duplicated sequence is plotted in orange. The height of each bar reflects the contribution of each outcome in the total amount of reads (*Y*-axis). **e** Clonogenic survival of the indicated mES cell lines after exposure to different doses of ionizing radiation. Data shown are the mean ± SEM (*n* = 6, three experiments using two independent clones per genotype). MF, mutation frequency; Delins, deletion insertion; snv, single nucleotide variant; TD, tandem duplication; TD+, tandem duplication with additional mutation (see "Methods" section); bp, base pair.

outcome in wild-type cells, however, strongly depends on NHEJ, resulting in a small fraction of tandem duplications (9.8% in wild-type cells versus 1.5% in $Ku80^{-/-}$ cells, Fig. 2b). These tandem duplications do not require the CST-complex, as knocking out $Ctc1$ did not alter the repair pattern on breaks with 5′ ssDNA protrusions (Fig. 2a–d). We previously found that excessive 5′ resection at Cas9-D10A-induced DSBs can lead to the exposure of 3′ protrusions that require TMEJ for repair[16]. We propose that CST's role in counteracting this excessive 5′ resection underlies the somewhat reduced mutation frequency in $Polq^{-/-}Ctc1^{-/-}$ cells (Fig. 2a), as well as the apparent greater proportion of TDs in the cognate mutation profile of double mutant cells (Fig. 2b): 5′ to 3′ end-resection removes the 5′ protrusions, and the outcome of subsequent repair will manifest as a deletion, not as a TD. A reduction of deletion outcomes is thus expected in $Polq^{-/-}Ctc1^{-/-}$ cells. How tandem duplications are formed at breaks with 5′ overhangs remains an unanswered question, although we previously found the NHEJ-polymerases Lambda and Mu to be involved[16].

The different contribution of CTC1 in the repair of Cas9-N863A-induced breaks (Fig. 1) as compared to Cas9-D10A-induced breaks (Fig. 2) is suggestive of a restricted role for CST in NHEJ, i.e. repair of DSBs with 3′ protrusions. To test this hypothesis in a more physiological setting, we analyzed the sensitivity of knockout cell lines toward ionizing radiation (IR), which is known to induce DSBs of different configurations[36]. In support of a confined function in stimulating NHEJ, we found CTC1 and STN1 deficient cells to be more sensitive than wild-type cells, however not as sensitive as cells lacking Ku80 (Fig. 2e and Supplementary Fig. 2b). In line with the mutagenic repair data, we found independent roles for TMEJ and CST in the processing of ionizing radiation-induced DNA damage: $Polq$-$Ctc1$ double knockout cells are more sensitive than either single-gene knockouts (Fig. 2e). Together, our data reveal that the CST complex is required to repair a specific subset of NHEJ substrates.

**Tandem duplication formation in human and mouse cells requires the 53BP1-Shieldin axis.** We next determined whether CST-mediated tandem duplication formation on 3′ ssDNA depends on the upstream NHEJ effectors 53BP1 and Shieldin, which can recruit CST to sites of DNA damage[21,37]. To this end, we generated knockouts for $Tp53bp1$ and for $Shld2$, the component of the Shieldin-complex that binds ssDNA through its OB-folds[22] (Supplementary Data 1, Supplementary Fig. 4a), in which we introduced DSBs with 3′ ssDNA overhangs in $eGFP$. Figure 3a shows a decreased mutation frequency upon loss of 53BP1 and Shld2, to a similar extent as observed upon loss of Ctc1 and Stn1 (Figs. 1d and S2c). Analysis of the mutational signatures further reveals an essential role for 53BP1 and Shld2 in generating most tandem duplications at DSBs with complementary 3′ ssDNA overhangs. The residual repair is characterized by microhomology usage, as was the case in CTC1 deficient cells (Figs. 3b, c, S4b compared to Fig. 1e, f). Similar results were obtained when targeting exon 3 of $HPRT$-$eGFP$: a strongly decreased mutation frequency and altered repair outcomes in $Tp53bp1$ and $Shld2$ knockout cells as compared to wild-type cells (Supplementary Fig. 5a–e). While all tested components of the NHEJ axis are equally required for mutagenic NHEJ on DSBs with 3′ protrusions, we found that 53BP1 separates from Shld2 and CST for the repair of DSBs with 5′ protrusions: whereas the mutational signatures in Shld2 and CST deficient cells marginally differ from that obtained from wild-type cells (Supplementary Figs. 5f–i, 2b–d), profiles in $Tp53bp1^{-/-}$ cells strongly deviate and are equally dependent on microhomology as those observed in Ku80 deficient cells (Supplementary Fig. 5h, i compared to Fig. 2c, d).

Still, in contrast to Ku80 deficient cells, TDs are formed at breaks with 5′ overhangs in the absence of 53BP1 with a similar frequency as in wild-type cells (Supplementary Fig. 5g).

Our data demonstrate that in mES cells 53BP1 and the Shieldin-complex are required for CST-dependent formation of small tandem duplications at DSBs with complementary 3′ ssDNA protrusions. We next addressed this biology in human cells by using REV7/MAD2L2 (a basal component of the Shieldin-complex[23,38,39]) deficient RPE1-p53$^{-/-}$ cells (Fig. 3d). As a target we first focused on the $FLT3$ locus; internal tandem duplications (ITD) within this oncogene are observed in a quarter of Acute Myeloid Leukemia (AML) cases[40]. We used different combinations of sgRNAs targeting the ITD-hotspot in $FLT3$ (Fig. 3e, Supplementary Fig. 6a). Targeted sequencing upon introduction of Cas9-N863A-induced DSBs at these sites (i) confirmed the robust formation of TDs at breaks with 3′ ssDNA in human cells (34.4% and 68.7% of the mutagenic events for sgRNAs L1 + R1 and L2 + R2, respectively) (Figs. 3f and S6b), (ii) revealed that those TDs have a strong resemblance with the ones found in AML patients (Fig. 3g)[2], and (iii) provided evidence for evolutionary conservation of the role of the Shieldin-complex in TD formation, as knocking-out REV7 drastically decreased the amount of TDs (Fig. 3f). To substantiate these conclusions we also targeted the human hemoglobin beta ($HBB$) locus, a clinically relevant target for gene-editing approaches, and obtained similar results (Supplementary Fig. 6c–e). We next wished to test whether also the human Shieldin-complex is specific to the repair of DSBs with 3′ ssDNA protrusions. To this end, we generated RPE1-p53$^{-/-}$ and RPE1-p53$^{-/-}$ REV7$^{-/-}$ cells stably co-expressing eGFP and either Cas9-N863A or Cas9-D10A (Supplementary Fig. 6f, g). Mutagenic repair of Cas9-induced DSBs was monitored by quantifying the ratio of eGFP negative over total cells upon delivery of sgRNA-duplexes that target the eGFP ORF at sequences identical to those targeted in the mES cell experiments; mutation profiles were derived by performing NGS analysis of the target site (Fig. 3h, i). Similar to mouse cells, we found that Shieldin deficiency in human cells strongly reduces the mutation frequency at Cas9-N863A-induced breaks and affects TD formation profoundly at breaks with 3′ ssDNA protrusions (compare N863A to D10A in Fig. 3h, i). Again, we verified these findings for another genomic locus, i.e., human HPRT exon 3 (Supplementary Fig. 6h–j).

**Tandem duplications are formed by Pol α-primase action at DNA break termini.** The CST-dependent fill-in we observe at complementary 3′ ssDNA overhangs can most easily be explained by the activity of Pol α-primase, a known interactor of the CST-complex[41]. Providing genetic evidence for this hypothesis is challenging because Pol α is essential for genome duplication, and we thus made use of a Pol α inhibitor (CD437)[21,42]. To minimize potential detrimental effects on cell viability, and prevent interference with normal DNA replication we designed an experimental setup in which we control the timing of DSB formation in cell populations synchronized in G2. To this end, we made use of wild-type $HPRT$-$eGFP$ mES cells stably expressing Cas9-N863A in combination with a regular and a light-inducible sgRNA targeting the eGFP site in $HPRT$-$eGFP$ (Fig. 4a and Supplementary Fig. 7a)[43]. After arresting cells in G2 using the CDK1 inhibitor RO-3306, we added CD437 to inhibit Pol α activity and one hour later induced DSBs with 3′ protruding ends at the $HPRT$-$eGFP$ locus by exposing cells to UV-light (Fig. 4b (timeline) and Supplementary Fig. 7b). Figure 4b shows that UV-induced TDs are readily detected six hours post-activation in mock-treated cells, but not in cells treated with CD437 (Fig. 4b), suggesting that Pol

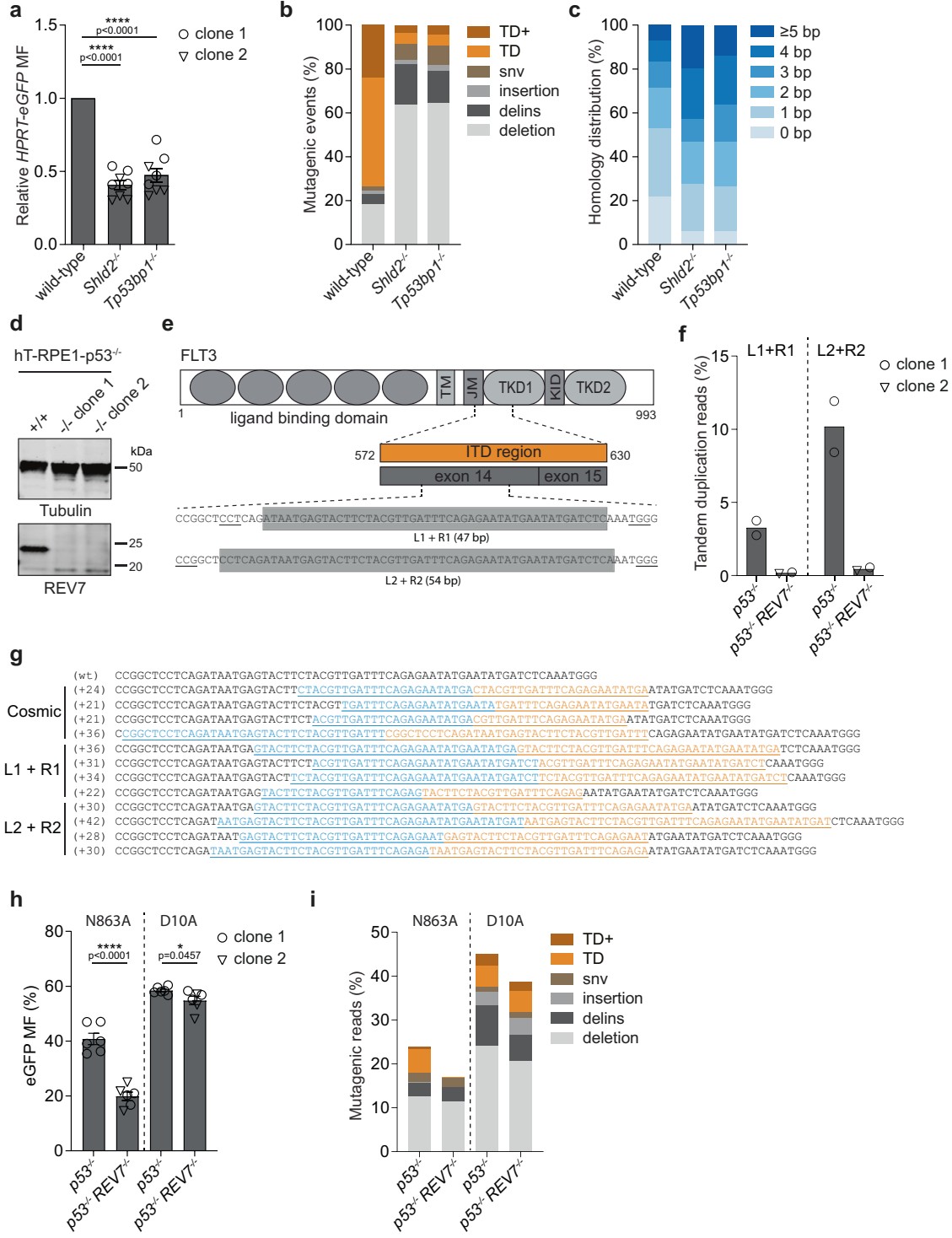

α-primase activity is required for TD formation at DSBs with complementary 3′ ssDNA protrusions.

To provide insight into the molecular mechanism of Pol α-primase activity at these breaks, we made use of one of its biochemical properties: eukaryotic primases exclusively use a purine as a cofactor to initiate RNA-primer synthesis opposite a pyrimidine[44,45]. This feature allows us to address primase action in vivo by adapting our experimental design to include DNA stretches exclusively made up of purines within the DSB overhangs, which prevents primer initiation. We hence call these stretches 'primase deserts'[46]. In the absence of such deserts, when

the tip is composed of random nucleotides, most TDs span almost the entire overhang, suggesting that fill-in initiates close to the terminus of 3′ ssDNA molecules (Fig. 4c, d, e).

We searched for genomic sites that upon targeting with Cas9-N863A results in DSBs with a primer desert at the tip of the 3′ protruding tail. We first chose two different combinations of Cas9-N863A target-sites located within a ~5 kb region on the X-chromosome to induce DSBs with 49 bp 3′ ssDNA overhangs: one containing a mix of purines and pyrimidines at the 3′ ends (ChrX random), the other containing a stretch of 15 purines at one 3′ end (ChrX PD). Compared to the random site, we found a

**Fig. 3 A requirement for the 53BP1-Shieldin axis in TD formation in human and mouse cells. a** Relative *HPRT-eGFP* mutation frequency for the indicated mES cell lines (using two independent clones per genotype) transfected with Cas9-N863A and sgRNAs targeting *eGFP*, producing DSBs with 3′ ssDNA overhangs. The data shown represent the mean ± SEM ($n = 6$) and are expressed as a fraction of the mutation frequency observed in wild-type cells (set to 1). Statistical significance was calculated via a two-tailed unpaired *t*-test. ****$P ≤ 0.0001$. **b** Quantification of the types of mutagenic events at breaks with 3′ ssDNA overhangs for the indicated mES cell-lines transfected with Cas9-N863A and sgRNAs targeting *eGFP*. GFP-negative (mutant) cells were sorted and used for targeted sequencing around the break-site, data represent the average of two independent clones per genotype. **c** Quantification of the extent of microhomology used for deletions and tandem duplications (TD) in the indicated genotypes. **d** Immunoblot confirming the loss of REV7 protein expression in two independent RPE1-p53$^{−/−}$ REV7 knockout clones (lower panel). An immunoblot for Tubulin is included as a loading control (upper panel). A representative example of at least two independent immunoblots is shown. **e** Targeting strategy of the human *FLT3* gene. The internal tandem duplication (ITD) region in exon 14 of the human *FLT3* gene was targeted using Cas9-N863A and two separate combinations of sgRNAs: L1 + R1 and L2 + R2, resulting in DSBs with 3′ overhangs of 47 bp and 54 bp, respectively; PAM-sites are underlined. **f** Quantification of the number of TD and TD+ reads derived from human RPE1-p53$^{−/−}$ and RPE1-p53$^{−/−}$ REV7$^{−/−}$ cells electroporated with Cas9-N863A and the indicated sgRNAs. Unselected cells were used for targeted sequencing around the break site. Data represents the average of RPE1-p53$^{−/−}$ samples and independent RPE1-p53$^{−/−}$ REV7$^{−/−}$ clones ($n = 2$). **g** Sequence representation of tandem duplication in the human *FLT3* gene found in the Cosmic database (top) and in RPE1-p53$^{−/−}$ cells transfected with Cas9-N863A and the indicated sgRNAs (bottom). The sequence that is duplicated is underlined in cyan and orange. **h** absolute eGFP mutation frequency for RPE1-p53$^{−/−}$ and RPE1-p53$^{−/−}$ REV7$^{−/−}$ cells stably expressing Cas9-N863A or Cas9-D10A transfected with sgRNAs targeting *eGFP*. The data shown represent the mean ± SEM ($n = 6$). Statistical significance was calculated via a two-tailed unpaired *t*-test. *$P ≤ 0.05$, ****$P ≤ 0.0001$. **i** Quantification of the number of mutagenic reads at breaks with 3′ ssDNA or 5′ ssDNA overhangs for the indicated RPE1 cell-lines expressing Cas9-N863A or Cas9-D10A, respectively, and transfected with sgRNAs targeting *eGFP*. Unselected cells were used for targeted sequencing around the break-site, data represent the average of four independent samples per genotype. MF, mutation frequency; Delins, deletion insertion; snv, single nucleotide variant; TD, tandem duplication; TD+, tandem duplication with additional mutation (see "Methods" section); bp, base pair.

---

profoundly different duplication profile in case the outermost 15 nucleotides in the tip are purines. The presence of the primase desert results in a clear depletion of events that are near full length, only at the overhang containing the purines (16.4% on the left overhang versus 82.6% on the right overhang) (Fig. 4f, g). TMEJ, in which Pol Θ uses terminal nucleotides of 3′ overhangs for microhomology-mediated repair, explains many of the residual TDs (Fig. S8a, b).

To exclude the possibility that the purine-stretch itself interferes with TD formation, e.g. due to secondary structure formation, we next analyzed tandem duplication formation by targeting genomic sites (on chromosome 3 and 8) such that the same primase desert is either at the tip or in the center of a similar-sized 3′ overhang. Consistent with the notion of Pol α-primase priming at the termini to explain CST-dependent fill-in we found that TD formation was only affected when the deserts are located at the tips; they did not affect tandem duplication formation when located ~15 bp from the outermost 3′ nucleotide (Figs. 4h, i, S8c–f). Of note, the biochemistry of NHEJ is not affected by terminal primase desert sequences, as DSBs with 5′ overhangs containing these sequences are as efficiently repaired by NHEJ as random sequences (Supplementary Fig. 8g: PD-tips versus HPRT exon 3). We also excluded a potential involvement for PrimPol (Supplementary Figs. 9a–S9c), another enzyme with primase activity[47].

## Discussion

Together, our study demonstrates that besides the recently described role in preventing the loss of genetic information flanking DSBs, the 53BP1-Shieldin-CST axis contributes to genome expansion by Pol α-primase mediated fill-in synthesis of complementary 3′ ssDNA protrusions, resulting in small tandem duplications. The induced overhangs in our study were as small as 43 bp, which may suggest that this length is sufficient for binding of Shieldin, however, we cannot exclude that 5′ end resection exposes longer stretches of 3′ ssDNA to facilitate binding. We found that TD size is dictated by the site where primase can initiate. Of interest, the fact that a substantial part of TDs encompass almost the entire 3′ ssDNA overhang, together with the realization that RNA primers are typically 10–15 nt long argues for a provocative idea that repair intermediates contain transiently embedded stretches of ribonucleotides, a feature recently proven to be important for NHEJ[48]. In addition, it

reveals that CST-dependent fill-in reaction on DSBs can be near-complete, leaving only a minimal 3′ ssDNA overhang of a few base-pairs to facilitate NHEJ (model Fig. 4j).

We here describe the biology that provides a plausible explanation for the etiology of sequence(repeat or homology)-independent genomic tandem duplication but also of the internal tandem duplications (ITDs) that are observed in several oncogenes (Fig. 3f–g)[2–5]. ITDs in the FLT3 gene, which we have here mimicked by introducing DSBs with 3′ protrusions, appear in radiation therapy-related AML and in mice exposed to ionizing radiation, arguing for mutagenic DSB repair at their origin[49]. This notion warrants further investigation into the use of specific CST and/or Pol α-inhibitors in combinatorial treatments, as to prevent therapy-induced activation of oncogenes.

## Methods

**Cell culture**. 129/Ola-derived IB10 mouse embryonic stem (mES) cells were cultured in mES knockout Dulbecco's modified Eagle's medium (Gibco) supplemented with 100 U/ml penicillin, 100 µg/ml streptomycin, 2 mM GlutaMAX, 1 mM sodium pyruvate, 1× non-essential amino acids, 100 µM β-mercaptoethanol (all from Gibco), 10% fetal calf serum and leukemia inhibitory factor. mES cell-lines were maintained on gelatin-coated plates containing irradiated primary mouse embryonic fibroblast feeder (MEFs) cells. For experiments, mES cells were cultured on gelatin-coated plates in Buffalo rat liver (BRL)-conditioned mES cell medium. Human hTERT-immortalized p53 deficient retina pigment epithelial (hTERT-RPE1-p53$^{−/−}$) cells were a gift from Rob Wolthuis and were cultured in DMEM (Thermo Fisher Scientific) supplemented with penicillin/streptomycin (Sigma) and 10% Fetal bovine serum (FBS, Bodinco BV)[50]. All cell lines were grown at 37 °C and 5% CO$_2$ and were frequently tested negative for mycoplasma contamination.

**Expression constructs**. Plasmid pU6-(BbsI)_CBh-Cas9-T2A-mCherry was a gift from Ralf Kuehn (Addgene plasmid #64324). Plasmids U6-Chimeric_BB-CBh-SpCas9n-D10A (PX335) and spCas9-N863A (PX856) were a gift from Feng Zhang (Addgene plasmid #42335, #62888, respectively). pU6-(BbsI)_CBh-Cas9-D10A-T2A-mCherry and pU6-(BbsI)_CBh-Cas9-N863A-T2A-mCherry were subcloned through ligation of an ApaI/AgeI-digested fragment of PX335 and an EcoRV/BsmI-digested fragment of PX856 into plasmid pU6-(BbsI)_CBh-Cas9-T2A-mCherry, respectively. To obtain Cas9-sgRNA expressing constructs, two complementary oligonucleotides (Integrated DNA Technologies) containing the target sequence and BbsI overhangs were phosphorylated, annealed, and cloned into the different BbsI digested pU6-(BbsI)_CBh-Cas9-T2A-mCherry plasmids[51]. An overview of the targeted sequences and their destination plasmids can be found in Supplementary Data 1 and S2. To obtain donor DNA for the tagging of the *HPRT* gene with eGFP, the mAID insert in pMK286-mAID-NeoR (a gift from Masato Kanemaki, Addgene plasmid #72824) was replaced with an eGFP PCR-product generated using primers containing BamHI and EcoRI recognition sites. Plasmids

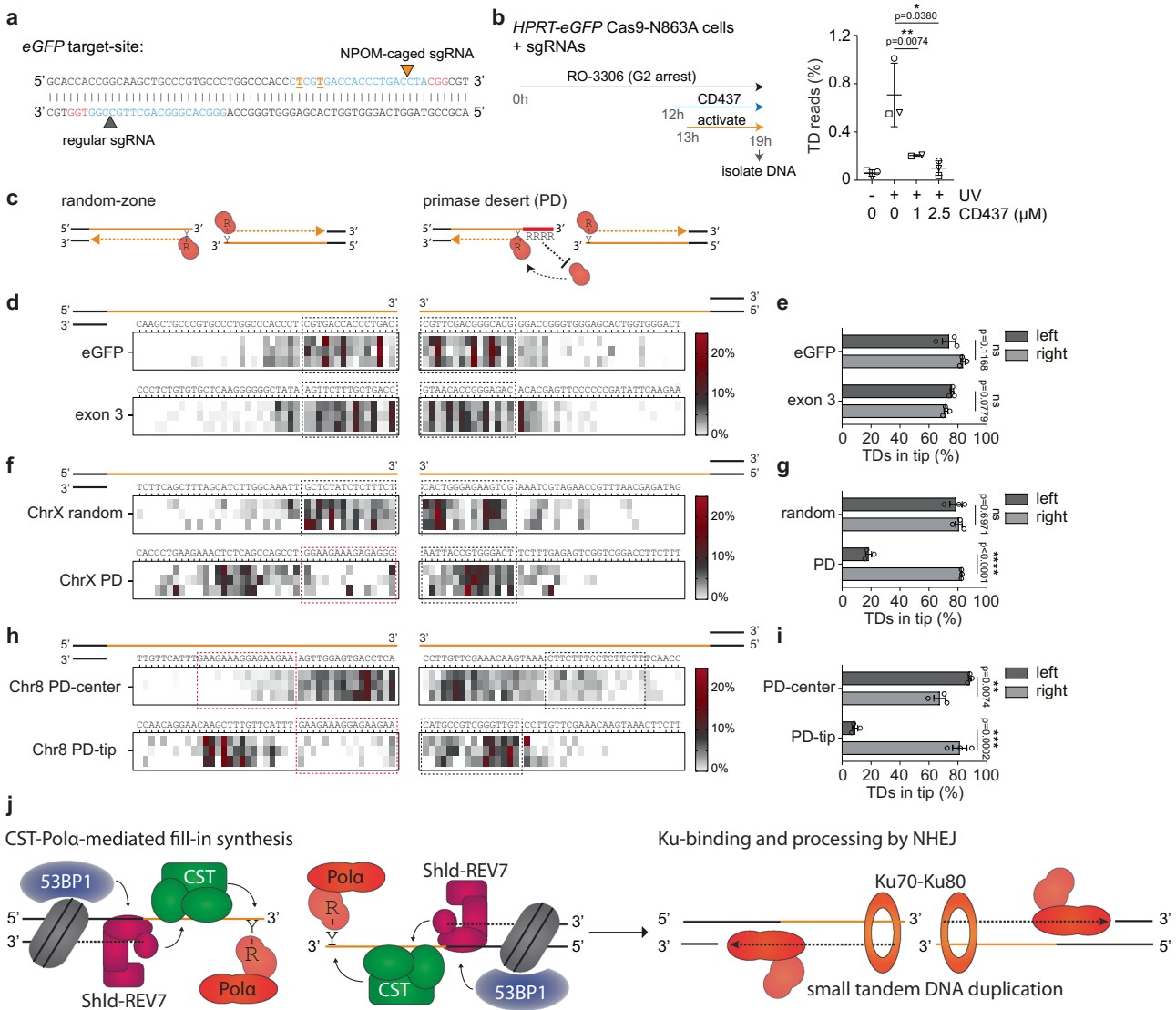

**Fig. 4 Pol α-primase action at the termini of DNA breaks explains CST-dependent TD formation. a** eGFP sequence with the sgRNA target-sites (in cyan) and PAM-sites (in red). The 6-nitropiperonyloxymethyl (NPOM) modified nucleotides in the right sgRNA are indicated in orange and underlined; theoretical sites of nicking by Cas9-N863A are indicated by triangles (light-induced in orange). **b** Quantification of the number of tandem duplication (TD) reads in control and CD437 treated mES cells (right) using the experimental timeline shown (left). Data shown represent the mean ± SD, independent experiments are indicated with different symbols. Statistical significance was calculated via a two-tailed paired $t$-test, $*P \leq 0.05$, $**P \leq 0.01$. **c** Graphic representation of the concept of fill-in synthesis at DSBs with 3′ protrusions mediated by primase. The right image illustrates how a stretch of purines, a primase-desert (PD), is hypothesized to prevent the start of fill-in synthesis, hence affecting the length of tandem duplications. **d** Heat maps representing the relative start position of TDs at 3′ overhangs as illustrated in **c** for Cas9-N863A-induced DSBs in *eGFP* (top) and *HPRT* exon 3 (bottom). **e** Quantification of the percentage of TDs from (**d**) that originate from fill-in synthesis starting in the tip (first 15 nucleotides from the 3′ end) of the overhangs of left and right DSB-end. **f** Heat maps representing the relative start position of TDs at 3′ overhangs for Cas9-N863A-induced DSBs at a random site (top) and at a site containing a PD in the tip of the left overhang (bottom) on chromosome X. **g** Quantification of the percentage of TDs from (**f**) that originate from fill-in synthesis starting in the tip (first 15 nucleotides from the 3′ end) of the overhangs of left and right DSB-end. **h** Heat maps representing the relative start position of TDs on the left and right 3′ overhang of Cas9-N863A-induced DSBs at a site containing a PD in the center of the overhang (top) or at the tip of the left overhang (bottom) on chromosome 8. **i** Quantification of the percentage of TDs from (**h**) that originate from fill-in synthesis starting in the tip (first 16 nucleotides from the 3′ end) of the overhangs of left and right DSB-end. All heat maps show three individual experiments per target site in wild-type mES cells and the data is shown as a percentage of the total amount of TDs in the spectra. Data shown in **e**, **g**, and **i** represent the mean ± SEM ($n = 3$), statistical significance was calculated via a two-tailed unpaired $t$-test. ns, not significant, $**P \leq 0.01$, $***P \leq 0.001$ $****P \leq 0.0001$. **j** Model for small tandem duplication formation. We propose that 53BP1-Shieldin-CST mediated fill-in synthesis of DSBs with 3′ overhangs by Pol α-primase leads to suitable substrates for Ku-dependent end-joining, hence resulting in small tandem duplications. Y, pyrimidine; R, purine; PD, Primase Desert; TD, tandem duplication.

pLenti-Cas9-N863A-2A-Blast and pLenti-Cas9-D10A-Blast were generated by PCR-amplification of Cas9-fragments from aforementioned Cas9-nickase constructs using primers containing BamHI and NheI recognition sites. PCR-products were cloned into the corresponding restriction sites of plasmid Lenti-Cas9-2A-Blast (a gift from Jason Moffat, Addgene plasmid # 73310).

**Transfections and nucleofection**. mES cells were trypsinized, counted, and resuspended in knockout Dulbecco's modified Eagle's medium (Gibco) and transfected in suspension using a Lipofectamine 2000 (Invitrogen):DNA ratio of 2.4:1. In general, $1.5 \times 10^6$ cells were transfected using 3 μg of total DNA and incubated for 30 min at 37 °C and 5% $CO_2$ in round-bottom tubes, subsequently

cells were seeded on gelatin-coated plates containing BRL-conditioned medium or irradiated MEFs. The medium was refreshed 24 h post-transfection. For the FLT3-targeting and HBB-targeting experiment, hTERT-RPE1 cells were electroporated using the Cell line Nucleofector Kit V and the Nucleofector 2b devices (both from Lonza). Per transfection, $7 \times 10^5$ cells were resuspended in 100 μl electroporation buffer (4 volumes 125 mM $Na_2HPO_4.7H_2O$, 12.5 mM KCL and 1 volume 55 mM $MgCl_2$) containing 5 μg of total DNA and transferred to cuvettes. Cells were electroporated using program T23 (Lonza) and subsequently grown in McCoy medium (Thermo Fisher Scientific) supplemented with 10% FBS on 6-wells plates for 24 h, after which medium was replaced for regular DMEM supplemented with penicillin/streptomycin and 10% FBS. For the use of sgRNA-duplexes, crRNAs (Supplementary Data 2) and tracrRNA were ordered at Integrated DNA Technologies (IDT) and annealed in Nuclease-Free Duplex Buffer according to the manufacturer's protocol. To target eGFP and HPRT exon 3 in hTERT-RPE1 cells, 0.6 μl of 10 μM sgRNA-duplexes (two times 0.3 μl for each sgRNA) and 0.9 μl of Lipofectamine RNAiMAX (Invitrogen) were mixed in optiMEM (Life Technologies) to a final volume of 50 μl, mixtures were incubated at RT for 20 min before adding to cells on a 96-wells plate (40.000 cells in 100 μl per well). The medium was refreshed 24 h post-transfection and cells were subcultured 48 h post-transfection. For the targeting experiment in mES cells with the Pol α-inhibitor, the volumes of the transfection mixture ingredients were extrapolated to transfect 400.000 cells in total on 35 mm dishes.

**Generation of cell lines**. To generate *HPRT-eGFP* knockin cell lines, ~1 kb homology-arms surrounding the *HPRT* stop-codon were amplified from genomic DNA using Phusion High-Fidelity DNA Polymerase (Thermo Fisher Scientific) and primers containing KpnI and SalI (left-arm) or NdeI and SalI (right-arm) restriction recognition sites. Digested PCR fragments were cloned into the *eGFP* donor-construct using restriction enzymes (NEB) recognizing the indicated restriction sites. Wild-type and previously described *Polq*−/− and *Ku80*−/− cells[16] were transfected with pU6-(BbsI)_CBh-Cas9-T2A-mCherry co-expressing a sgRNA that targets the stop-codon of *HPRT* and the plasmid containing the *HPRT* homology-arms in a 1:1 ratio (Supplementary Fig. 1A). One week after transfection, GFP positive cells were sorted using a BD FACSAria (BD Biosciences) and seeded at low density and maintained with regular medium changes until colonies formed, after which multiple clones per cell line were picked and expanded. *HPRT*-targeting was confirmed by PCRs specific for the left-border and right-border of the targeted region using PrimeSTAR GXL DNA Polymerase (Takara Bio) (Supplementary Fig. 1A). Two independent clones per genotype were selected and used in this study.

*Ctc1*, *Stn1*, *Shld2*, and *Tp53bp1* single knockout cell lines were generated by transfecting wild-type *HPRT-eGFP* mES cells with plasmids co-expressing Cas9-WT-2A-mCherry and the corresponding sgRNA. Two independent sgRNAs per gene were used to transfect *HPRT-eGFP* wild-type clone 1 or clone 2, respectively. *Polq-Ctc1* double-knockout cell lines were generated by targeting the *Ctc1* gene in *HPRT-eGFP Polq*−/− clone 1. *PrimPol* single knockout cell lines were generated in wild-type mES cells. Two days after transfection, mCherry positive cells were sorted and seeded at low density and maintained with regular medium changes until colonies formed, after which multiple clones per transfection were picked and expanded. Clones with bi-allelic mutations were identified using restriction fragment length polymorphism (RFLP) of specific PCR products[52] and by Sanger-sequencing of the targeted region. One clone per sgRNA was selected for further use, resulting in two independent clones per genotype. An overview of the targeted sequences, the primers used for PCRs, the restriction sites used for RFLP, and the introduced bi-allelic mutations can be found in Supplementary Data 1.

Human REV7 knockout cell lines were generated by transfecting hTERT-RPE1-*p53*−/− cells with a plasmid co-expressing Cas9-WT-2A-GFP and a sgRNA targeting *REV7* (TGTGCTCTGCGAGTTCCTGG) using Lipofectamine 2000. Two days after transfection, GFP-positive cells were sorted and seeded at low density and maintained with regular medium changes until colonies formed, after which multiple clones were picked and expanded. Knockout of REV7 in individual clones was confirmed by immunoblotting.

eGFP positive hTERT-RPE1 were generated by lentiviral transduction with empty pLentiGuide-GFP-2A-Puro (a gift from Sylvie Noordermeer) and by culturing cells subsequently in a medium supplemented with 1.5 μg/ml Puromycin (Gibco). hTERT-RPE1 or mES cells stably expressing Cas9 were generated by lentiviral transduction with pLenti-Cas9-N863A-2A-Blast or pLenti-Cas9-D10A-Blast and by culturing cells subsequently in a medium supplemented with 5 μg/ml Blasticidin (Gibco).

**Immunoblotting**. Cells were lysed in 2% SDS, 1% NP-40, 50 mM Tris pH 7.5, and 150 mM NaCl. Protein-lysates were equalized based on protein concentration measured using the BCA Protein Assay Reagent (Thermo Fisher Scientific). Protein samples were separated on Novex 4–12% Bis-Tris gradient gels using MOPS SDS running buffer and NuPage LDS sample buffer (all Thermo Fisher Scientific); subsequently, proteins were transferred onto Immobilon-FL membranes (Merck Millipore). The following primary antibodies were used: Anti-PARP1 (rabbit polyclonal, Cell Signalling Technology #9542, 1:1000), anti-OBFC1 (STN1, mouse monoclonal, Santa Cruz sc-376450, 1:1000), anti-53BP1 (rabbit polyclonal, NOVUS biologicals NB100-304, 1:1000), anti-Mad2L2/REV7 (rabbit monoclonal,

Abcam ab180579, 1:1000), anti-Cas9 (mouse monoclonal, 7A9-3A3 Cell Signaling Technology, 1:1000) and anti-Tubulin (mouse monoclonal, Sigma-Aldrich, T6199 clone DM1A, 1:5000). The secondary antibodies CF680 goat anti-rabbit IgG and CF770 goat anti-mouse IgG (Biotium, both at 1:10.000) and the Odyssey CLx infrared imaging scanning system (LI-COR Biosciences) were used to detect protein expression.

**Cell survival assay after ionizing radiation (IR)**. Cells were trypsinized, counted, and seeded at low density on culture dishes (60 mm). Cells were exposed to IR using a YXlon X-ray generator (YXlon International) and left to grow for 7–9 days. Subsequently, cells were washed with 0.9% NaCl and stained with methylene blue to score the number of surviving colonies. The survival of treated cells was calculated relative to the cloning efficiency of untreated cell lines (0 Gy), which was set to 1.0 for each individual genotype.

**HPRT-eGFP and eGFP mutation assay**. Cells were transfected with two Cas9-Nickase-2A-mCherry constructs in a 1:1 ratio, each expressing one of the two sgRNAs targeting closely located sequences on opposite strands in the *HPRT-eGFP* gene. Two days after transfection cells were passaged and a fraction was used to determine the transfection efficiency by measuring the percentage of mCherry-positive cells on a NovoCyte flow cytometer (ACEA Biosciences). Subsequently, the mutation frequency was analyzed 7 days post-transfection by measuring the percentage of GFP-negative cells on the NovoCyte using the NovoExpress software (version 1.4.1). The absolute mutation frequency was calculated by correcting for the transfection efficiency measured on day 2. To measure the eGFP mutation frequency in RPE1-hTERT cells, cells stably expressing Cas9-N863A or Cas9-D10A were transfected with sgRNA-duplexes targeting eGFP. Two days after transfection cells were passaged, subsequently the mutation frequency was analyzed 6 days post-transfection by measuring the percentage of GFP-negative cells.

**Pol α-inhibitor experiment**. *HPRT-eGFP* wild-type mES cells stably expressing Cas9-N863A were transfected with sgRNA-duplexes and seeded on 35 mm dishes (400.000 cells per transfection). Subsequently, 10 μM RO-3306 (Sigma-Aldrich) or equal volumes of DMSO were added to the cells. After 12 h, the Pol α-inhibitor CD437 (Sigma-Aldrich) or equal volumes of DMSO were added to the cells. One hour after the addition of CD437, the 6-nitropiperonyloxymethyl (NPOM)-modified sgRNA was activated by exposing cells for 20 min to a Chromato-Vue TL-33 UV Transilluminator (UVP) that delivered 3 mW/cm² of 290–390 nm wavelength light. Six hours after activation, cells were harvested and one fraction was used for DNA isolation the other fraction was used to fix cells and to stain DNA using propidium iodide (Sigma-Aldrich).

**Targeted sequencing of Cas9-induced repair outcomes**. HPRT-mutant cells were selected by sorting ≥100.000 GFP-negative cells on a BD FACSAria III using BD FACSDiva software (version 9.0.1, BD Biosciences) (an example of the gating strategy can be found in the source data file) or by seeding 500.000 cells in 6-thioguanine containing medium 7 days post-transfection as indicated and allowed to grow for 5–7 days. For the *FLT3*-, HBB-, eGFP- and HPRT exon 3-targeting in RPE1-hTERT cells and the primase desert experiments in mES cells, unselected cells were harvested 5–6 days post electroporation/transfection. Genomic DNA was isolated by lysing cell pellets into 10 mM Tris-HCL pH 7.5, 10 mM EDTA, 200 mM NaCl, 1% SDS, and 0.4 mg/ml Proteinase K. Lysates were incubated at 55 °C for 3–16 h after which the lysis was neutralized by adding saturated NaCl and centrifugation at $20,000 \times g$ for 10 min. DNA was precipitated and extracted by adding one volume of isopropanol to the supernatant followed by centrifugation, washing with 70% ethanol, and resuspension in TE.

For sequencing, primers specific for the targeted regions have been selected that yield a ~150–200 bp product on wild-type alleles and that contain adaptors for the p5 and p7 index primers (5′-GATGTGTATAAGAGACAG-3′ and 5′-CGTGTGCTCTTCCGATCT-3′, respectively). These primers were used to amplify the targeted region using Phusion High-Fidelity DNA Polymerase (Thermo Fisher Scientific) and the following conditions: 95 °C for 10 min, 25 cycles of 95 °C for 10 s, 60 °C for 30 s, and 72 °C for 30 s, and the final extension 72 °C for 5 min. The PCR products were purified using a 1.8× reaction volume of magnetic AMPure XP beads (Beckman Coulter) according to the manufacturer's protocol and DNA was eluted in 20 μl MQ. Flow-cell adaptor sequences were added by performing PCRs with 5 μl purified PCR-product, 0.3 μM of p5 and p7 index primers and Phusion High-Fidelity DNA polymerase and the following conditions: 95 °C for 10 min, 5 cycles of 95 °C for 10 s, 58 °C for 30 s and 72 °C for 30 s, and the final extension 72 °C for 5 min. The PCR products were purified with AMPure XP beads and eluted in 20 μl MQ. DNA concentrations were measured using the Quant-iT dsDNA assay kit and the Qubit Fluorometer (both Thermo Fisher Scientific) according to the manufacturer protocol and samples were pooled at equimolar concentrations per target-specific PCR. The quality and quantity of these pools were analyzed using a High Sensitivity DNA kit and Bioanalyzer (Agilent) which was used to generate an equimolar library that was sequenced on a NovaSeq6000 (Illumina) by 150-bp paired-end sequencing or on a MiSeq (Illumina) by 300 bp paired-end sequencing (Supplementary Data 4).

**Sequence analysis of Cas9-induced repair outcomes**. To map raw NGS sequences to a reference sequence, paired-end sequence files were first assembled using FLASH2 (v2.2.00) with parameters: -M 5000 -O -x 0. Using a custom JAVA program (manuscript in preparation, available upon request) reads were filtered and aligned to a reference sequence containing the primer sequences and the CRISPR-Cas9 target sites. Bases with error probability >0.05 were masked to obtain high-quality sequences. Only reads where both the start and end of the sequence lies within the specified primers were kept. In addition, events had to start >5 bp from both primers to ensure primers were annealed at the primer sites in the reference. The reads that passed these filters were classified into the following groups: deletion, insertion, delins (deletion with insertion of de novo DNA at the location of the deletion), snv (single-nucleotide variant), wild-type (i.e., identical to the reference sequence), tandem duplication (TD, insertion of ≥6 bp were the exact sequence is found immediately adjacent to the location of the insertion), tandem duplication compound (TD+, tandem duplication with additional mutation; these products can result from consecutive rounds of nicking and repair upon TD-formation, as this does not eliminate the sgRNA targets[16]. wild-type reads were excluded from analysis, all data in the figures (except for unselected samples in the RPE1-hTERT targeting experiments and in the experiments with Pol α-inhibitor) is analyzed relative to the total amount of mutant reads. Of note, the 'snv' class potentially also includes errors introduced on wild-type products during the PCR, especially in unselected samples. Additional parameters such as deletion/insertion size and microhomology length for deletions and TDs were determined for each event. Finally, R (version 4.0.2) was used to generate plots of the mutational spectra for each genotype.

**Statistical analysis**. GraphPad Prism software (8.4.2) was used for statistical analysis. For mutation frequencies, a two-tailed unpaired t-test was used; replicate number, mean, error bars and P values are explained in the figure legends. For mutational signatures and distribution of homology, Two-Way ANOVA with recommended correction for multiple comparisons was used; test details and adjusted P values can be found in Supplementary Data 3.

**Reporting summary**. Further information on research design is available in the Nature Research Reporting Summary linked to this article.

## Data availability

The raw targeted sequencing data generated in this study have been deposited in the NCBI SRA database under accession code PRJNA641538. A description of the sequence files can be found in Supplementary Data 4. Source data are provided with this paper. All data is available from the authors upon reasonable request. Source data are provided with this paper.

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

## Acknowledgements

We thank Dr. M.S. Luijsterburg and Annelot Wondergem for generating the RPE1-p53$^{-/-}$REV7$^{-/-}$ cell lines and Marco Barazas for generating the pLenti-Cas9-Nickase-2A-Blast constructs and for critical reading of the manuscript. This work was supported by a Young Investigator Grant from the Dutch Cancer Society (KWF, 2020-1/12925), a Veni-grant (016.Veni.171.028) from The Netherlands Organization for Scientific Research for Earth and Life Sciences to J.S., and grants from the Dutch Cancer Society (11251/2017-2) and the Holland Proton Therapy Centre (2019020-PROTON-DDR) to M.T.

## Author contributions

J.S. and M.T. conceptualized the study and designed experiments. J.S., N.M.S., and H.K. generated mES knockout cell lines, performed Cas9-targeting experiments, and generated NGS samples. N.M.S. and H.K. performed IR-survival experiments. R.v.S. wrote the custom Sequence Interrogation and Quantification program, analyzed sequence data, and selected genomic regions for the primase desert studies. J.S. analyzed the data; J.S. and M.T. interpreted the results and wrote the manuscript with input from all co-authors.

## Competing interests

The authors declare no competing interests.
