## [Peer Review File · Nature Communications]

We would like to thank all reviewers for their time and their constructive comments and suggestions. We feel that the outcomes of the proposed experiments have significantly strengthened our manuscript and provide further support for the presented model. Also, the suggestions for textual changes and additions helped us to increase the readability of our manuscript.

Reviewer #1 (Remarks to the Author):

In this manuscript, Schimmel et al. describe a novel role for DNA polymerase- α primase in the generation of tandem duplications (TD). Using CRISPR-Cas9 system in mouse ES cells, the authors have generated double strand breaks (DSBs) with 3' or 5' overhangs in exon 3 of HPRT gene and a eGFP reporter targeted to exon 9 of the same gene. ES cells with loss of HPRT or eGFP function were used to PCR amplify and sequence the region around the site of CRISPR-mediated DSBs. ES cells with mutation in genes of interest were used to determine their role in TD generation. The findings revealed that 53BP1, components of the CST complex (CTC1, STN1) and the Shieldin complex (Shld2, REV7) play a critical role in TD generation. Loss of any of these suppressed TDs at DSBs with 3' overhangs. In contrast, loss of DNA pol theta, required for Theta mediated end joining (TMEJ) was found not to be involved in the generation of TDs with this process. Role of REV7 was validated in generation of TD in FLT3 gene human RPE1 cells .

Overall the novel findings described in the study are important and they reveal a novel role for the 53BP1-Shieldin-CST complex and Pola primase in the generation of TD. The experiments are well designed with proper controls. The only major concern is that most of the studies are performed only in mouse ES cells. The authors have used human RPE1 cells to validate the role of REV7 in the generation of TDs in FLT3 gene, which is known to be prone to TDs in patients with radiation-therapy related AML. It is important to validate some of the key findings in RPE1 cells at some locus other than FLT3. This concern stems from the mild reduction in TDs in exon 3 in Ku80^{-/-}, Cst1^{-/-} and Stn1^{-/-} cells (figure S3).

We have now included data from three additional target-sites in RPE1 cells, which together demonstrate that the presented biology for mES cells is also found in human retina cells:

- I. We targeted the HBB-locus, where TD-formation in human cells at Cas9-N863A induced breaks has been previously reported¹. These data are now presented in Fig S6C-E.
- II. We created eGFP-positive RPE1 cells to be able to target the exact same eGFP-sites as we used in mES cells, which also enabled us to determine the GFP mutation-frequency in those cells. The results are now included in Figure 3 (3H-3I) and in supplemental Figure 6 (S6F-G).
- III. We targeted the human HPRT exon 3 site, which shows strong homology to the site that was targeted in mES cells (Fig S6H-J).

For the latter two target sites we also performed experiments with Cas9-D10A (Fig 3H-I, Fig S6I-J).

These data reveal a strong reduction in the amount of TDs formed at Cas9-N863A-induced breaks (DSBs with 3' overhangs) in REV7 knockout cells at all three additional sites, confirming the previously results at the FLT3-locus. Also, confirming mES cell data, human Shieldin has a marginal role in the mutagenic repair of DSBs with 5' overhangs.

With regard to the reviewers conclusion that the reduction in TDs at HPRT exon 3 is mild, it may be important to note that the histograms in e.g. figure S3b (or figure 1e) are set to 100% and reflect the

mutational profile, but NOT the frequency, which is presented in *e.g.* figure S3a. In the case of the HPRT exon 3, the mutation frequency in *Ku80^{-/-}* and *Ctc1^{-/-}* is 20-25% of that in wild-type. Within that 75-80% reduced population the TD percentage goes down even further. Thus, the absolute amount of TDs in the mutant cells is strongly decreased as compared to wild-type cells. In addition, the molecular parameters for TDs that are found in the knockout cells (*e.g.* the degree of microhomology present at the junctions (figure S3c)) are different than for the majority of TDs found in wild-type cells, further supporting the conclusion that also on this DSB site, Ku80 and CST-dependent TD-formation is the dominant mechanism.

We have had a lot of discussion in our lab on how to best visualize the data to avoid (this very understandable) confusion but feel that the chosen visualization best represent the actual data without subjective interpretation. For instance, a potentially more intuitive measure for the drop in TD frequency is to take the product of the mutation frequency of the target and the TD ratio of the mutant population. However, this is incorrect as it would assume that we capture all possible mutations in the NGS assay.

We have now included text in the result section to make this more clear.

Minor comments:

1. Use of “HPRT mutation frequency” to label the y-axis in Figures 1C and S2C and 3A is incorrect. These cells were not selected for HPRT loss i.e. they are not 6-TG resistant cells. In these figures, y-axis should be labeled as “GFP mutation frequency”.

Indeed, the labeling of these y-axes was incorrect and we thank the reviewer for pointing this out. Our analyses are based on an *HPRT-eGFP* fusion gene, where a single GFP-copy was inserted in-frame with the last exon of *HPRT*. This allows us to measure mutagenic repair of Cas9-induced breaks introduced throughout the whole gene by quantifying the amount of GFP negative cells, so also upon targeting exon 3 of the *HPRT* ORF as used in Figure 2, S3 and S5. For this reason and for consistency, we have now changed the y-axis label of all these figures to ‘Relative *HPRT-eGFP* MF’. We now also added a sentence at the first paragraph of the ‘results’ section and modified Figure 1 to better explain the generation of the *HPRT-eGFP* tagged gene.

2. Related to the above point, in Figure 1B, HPRT mutants should be described as “6-TG resistant or GFP negative”. The way it is written, it is confusing as it gives the impression that the cells are 6-TG resistant and GFP negative. As explained in the methods section, eGFP targeted cells were selected for GFP loss and exon 3 targeted cells were selected for 6-TG resistance.

We now changed the figure (old Figure 1B to new Figure 1A) to more clearly capture the essence of the assay. To elaborate on this issue: throughout the manuscript we target two different sites in the *HPRT-eGFP* fusion gene: one in exon 3 of the *HPRT* coding sequence and one in the eGFP coding sequence. We observed that targeting both sites leads to both 6-TG resistant and GFP negative cells. To determine the mutational profiles, we used sorting of GFP-negative cells for most experiments, also for experiments where exon 3 was targeted. The only exception is the experiment described in Figure S5 (exon 3 targeting in *53bp1* and *Shld2* deficient cells) where we had to use 6-TG selection, due to Covid-related limited availability of our sorting facility at the time we were running those experiments. This is why we mention both selection methods in Figure 1B and the Methods section. The selection method used for each individual experiment is described in the figure legends.

For the exon 3 target site we do not observe significant differences in mutational signatures between 6-TG or GFP-negative selected samples, as shown for data from wild-type cells below:

3. Figure S2 is described prior to Figure S1 in the text.

This has been changed

4. Figure S2A does not match with text. There is no figure showing Ctc1 knock-out generation.

This has been changed. Figure S2A refers to the STN1 knock-out generation.

5. It is unclear why *Polq*^{-/-} *Ctc1*^{-/-} double knockout cells show clearly more TD in the repair of 5' ss overhang (Fig 2A).

We suspect that also here the way the data are presented may lead to some confusion. It is indeed true that in the *Polq*^{-/-} *Ctc1*^{-/-} double knockout cells, the relative contribution of TDs increases. However, the absolute frequency of mutation in the double knockouts is ~ 35% of that of wild-type cells. It may thus be argued that the absolute frequency of TDs is not affected (~30% of ~0.35 MF versus ~10% of ~1.00 MF).

This reasoning nevertheless defers the question to something that is also unclear to us: the observation of a lower mutation frequency in *Polq*^{-/-} *Ctc1*^{-/-} double knockout cells compared to *Polq*^{-/-} cells, which argues that *Ctc1* also has some suppressor activity on mutation induction in *Polq*^{-/-} cells at 5' overhangs. At present, we can only speculate: one, to us reasonable, explanation may be that for a subfraction of DSBs (in yet to be defined cellular contexts, e.g. cell cycle phase), the 5' protruding end is subject to (excessive) end-resection, which can be countered by *Ctc1*-mediated fill-in of the newly arisen 3' overhang followed by NHEJ. Because the end-resection removes the overhangs, this outcome will not manifest as a TD but as a deletion in a mutation profile. In the absence of TMEJ (*Polq*^{-/-}) and of *Ctc1*-mediated fill-in/NHEJ it may be that specifically these DSBs are unreparable (CST-mediated fill-in is necessary for 3' overhang repair by NHEJ and TMEJ is necessary for 3' overhang repair by TMEJ) and cells containing these DSBs either die or at least fail to proliferate, hence dropping out of our analysis. This would explain both a decreased mutation frequency as well as a relatively more affected drop in deletions (giving the impression of more TDs) for double mutant cells. This possibility has now been mentioned in the result-section as a potential explanation for the observed spectrum in those cells (lines 118-122)*.

6. The epistatic relationship between CST and Ku80^{-/-} described in the text should be explained clearly. It is unclear how the conclusion was drawn based on the results shown in Fig 1C.

This statement was based on the fact that similar to the situation in Ku80 deficient cells (Schimmel et al. EMBO, 2017, Figure 3B)² mutation induction at Cas9-N863A-induced breaks in *Ctc1* knockout cells completely relies on TMEJ. We, however, realize that such a statement would require experimental validation. We made several attempts to generate *Ku80-Ctc1* double knockouts, but unfortunately did not manage to get viable clones homozygous for both genes, probably because these genes also have independent roles in maintaining genome stability (e.g. at telomeres). We therefore amended the text to be correct.

* Line numbers correspond to the manuscript with all changes accepted.

Reviewer #2 (Remarks to the Author):

The authors previously identified an association between tandem duplication and double strand breaks containing long complementary overhangs. Here they identify a role for the CST complex, and potentially primase, as the mechanism behind these TDs when they are associated with 3' overhangs. The authors make a compelling case for the significance of their observation; a nice experiment at the *Flt3* locus shows it could explain AML-associated tandem duplications in this gene. There are also some well-performed experiments differentiating the roles of 53BP1, MadL2, the CST complex, and the two end joining pathways when considering the overhangs with different strand polarity. There is a lot of important information in a small package.

An elegant experiment described in Figure 4 makes a strong case for a mechanism requiring primase-dependent fill-in of the 3' overhang, but also represents the only significant concern regarding this work. It is difficult to conceive of a primase-independent mechanism by which a pyrimidine-rich strand would alter the distribution of tandem duplications (a role for PrimPol, which may also be dependent on a pyrimidine in the template? The pyrimidine rich region may form a secondary structure that blocks activity of a synthetic enzyme other than primase or PrimPol?). Regardless, the authors did not use genetic methods or an inhibitor to directly link primase or Pol alpha activity to TD formation, making the title "...result from CST-guided Primase-Pol alpha action..." somewhat of an over-statement. Use of Pol alpha inhibitors (like the two used in Mirman et al.) could help address this concern, and also help support the presumptive link between the observations in these two papers. Alternatively, different language like "consistent with a requirement for primase activity" would resolve the concern.

We agree with this reviewer that the use of Pol-alpha inhibitors in our experiments would further strengthen our manuscript. Prior to our first submission we carried out this type of experiments but it had been challenging to time stringent yet sub-lethal concentrations of Pol-alpha-inhibition together with Cas9-break induction without interfering with proliferation/viability of the cells. We have resolved this issue by using a light-inducible Cas9-approach³, which we have used in newly added experiments to induce breaks with 3' ssDNA overhangs in cells arrested in the G2-phase in the presence or absence of the Pol-alpha inhibitor CD437. This experimental setup reveals a clear

reduction in TD formation upon inhibition of Pol-alpha (now included as Figure 4A-B and Figure S7), which provides additional proof for the involvement of this polymerase.

Additionally, we excluded a role for PrimPol in TD-formation. We generated *PrimPol* knockout cell-lines for which we established the mutational profile of Cas9-N863A-induced breaks at *HPRT* exon 3. The frequency of TD formation, their characteristics (size and homology) and the site of initiation is not affected in those cells as compared to wild-type cells. These data are now included in Fig S9).

Minor suggestions.

Line 99. "We found that TMEJ and NHEJ can act almost completely redundant on these types of breaks (Figure 2A).9" "almost" and "completely" are contradictory, and must be resolved with a single modifier. Consider replacing both words with "largely".

We thank the reviewer for this suggestion; the text has been amended.

Line 106. "How tandem duplications are formed at breaks with 5' overhangs remains an unanswered question, but was previously shown to require the action of the NHEJ-polymerases Lambda and Mu 9. "...remains unanswered..." and "...was previously shown to require..." is at best confusing. The cited work also shows that TD frequencies are only modestly reduced after loss of both genes, arguing "require" is not accurate.

We have now changed this to: "How tandem duplications are formed at breaks with 5' overhangs remains an unanswered question, although we previously found an involvement of the NHEJ-polymerases Lambda and Mu.

The way the tandem duplications were represented in Figure 3F was initially confusing, since while the intention is to represent insertions, the sequences were identical through all rows. For the mutant sequences, consider showing only inserted (tandemly duplicated) sequence, rather than coloring it differently, and represent identity in flanking DNA with hyphens.

We have now amended this figure (new Fig 3G)

* Line numbers correspond to the manuscript with all changes accepted.

Reviewer #3 (Remarks to the Author):

In this manuscript the Authors tackle some aspects of the mechanism underlying the generation of tandem duplications at double-strand breaks. In a previous publication (Schimmel et al., EMBO J., 2017; ref.9) the Authors found that tandem duplications form at DSBs with small 3' protruding end. They now expand on these findings and demonstrate that DNA polymerase-alpha-Primase exerts a crucial role in this process, alongside the 53BP1-CST-Shieldin complex. Additionally, they show that this mechanism acts only on DNA breaks presenting with 3' protruding ends.

The main concern that we have relates to the novelty. The localization of Shieldin, in a 53BP1

dependent manner at DSBs has already been elegantly demonstrated as well as the role of this complex in protecting DNA free ends and avoiding hyper-resection (Noordermeer et al., Nature, 2018; ref.15). Also, the role exerted by the CST complex in DNA repair, beyond the protection of telomeres, through the interaction with Shieldin complex has been shown by Barazas et al., Cell Reports 2017, ref.17. Additional reports have shown how the CST complex mediates a fill-in reaction driven by polymerase alpha both at telomeres and DSBs (Miyake et al., Mol. Cell, 2009; ref.25; Mirman et al., Nature, 2018; ref. 14). Also, the engagement of the complex described in this manuscript has been reported in other, similar settings, such as telomeres. As such, we are not convinced of the novelty and the impact of the story presented here.

While we recognize the profound importance of the previously published work for our study, and also a certain degree of conceptual overlap, we feel that our work goes beyond the state of the art on three important issues: i) the experimental demonstration at nucleotide resolution of fill-in, and all the way up to the end, using the primase desert novelty we introduced, provides an explanation for how DSBs with 3' protruding can be made into a substrate for NHEJ. This is new mechanistic insight as currently proposed models (in the papers mentioned above) point in very different directions (e.g. nucleolytic removal of larger >50bp overhangs). ii) we demonstrate substrate specificity (DSBs with 3' and not 5' protruding tails) for this EJ axis, providing an explanation for the only mild sensitivity of mutant cells to ionizing radiation. iii) we demonstrate that (and how) this EJ axis underlies a specific mutagenic outcome at DSBs.

Specific points:

1) In the assay reported in Figure 1 the impairment of TMEJ and NHEJ activity (that is, the double mutant Polq-Ctc1) reduces the mutational frequency. The Authors should discuss the meaning, and the potential implications of these results. Does it mean that cells repair the lesion with high fidelity, or that the cell lose the corresponding DNA sequences? Although it is an established assay, given the biology described, the Authors should describe, and in the test explain the results obtained and its implications.

We agree with the reviewer that this was insufficiently explained in the text. We now added a section to better explain the assay (lines 59-65)* and also added more text on the implications, e.g. on whether DSBs in double mutant cells are rerouted to error-free repair or are dropping out of the analyses (lines 78-80)*.

2) In Figure 1C the Authors show that Ku80 and Ctc1 knockout clones cause a very similar reduction in mutation frequency, concluding that "CST-complex is required to repair DSBs with 3' overhangs presumably due to its role in NHEJ". Although it has been shown in the literature that CST-complex interacts with 53BP1 in the NHEJ pathway, it cannot be concluded that it interferes with mutational frequency across the NHEJ. To conclusively support this statement, the authors should investigate how Ku80-Ctc1 double knockout clones impact the mutational frequency, if they decreases it to levels similar to those of single knockout clones or if they completely abolish it like the Polq-Ctc1 double knockout clones.

We agree with the reviewer that the mutation frequency data by itself does not provide sufficient evidence for this statement. The experimental evidence for nonfunctional NHEJ on DSBs with 3' overhangs in absence of CST comes from the observation that *Ctc1* or *Stn1* knockout cells fail to produce NHEJ repair outcomes on this type of breaks (which was discussed later). The repair profile perfectly matches that of *Ku80*^{-/-} cells and displays all the characteristics of a TMEJ spectrum. We have now amended the text to correct for the premature mentioning of this conclusion.

In an attempt to experimentally address epistasis, we made repeated attempts to generate *Ku80-Ctc1* double knockouts, but unfortunately did not manage to get viable clones that were homozygous for both genes (while being prolific in making knockouts and also obtaining large numbers of heterozygotes). This outcome is likely because the encoded proteins also have independent additional roles in maintaining genome stability (e.g. at telomeres). This means that the phrasing on a potential epistatic relationship between CST and Ku80 is too strong and we thus decided to remove this sentence.

3) In Figure 1D the Authors correlate the reduced mutational frequency of clones with their ability to survive. However, the mutational frequency is calculated in response to the formation of DSBs with 3' ssDNA protrusions obtained with the as9-N863A system, while survival is measured in response to DSBs produced following exposure to ionizing radiation. Are the two systems comparable? Ionizing radiations do not produce DSBs with 3' ssDNA protrusions, or certainly not only. How can these two data be compared? Are there differences between 3' ends generated by Cas9 system from those generated by ionizing radiation? Could these differences explain some discrepancies between the results presented in Figure 1C and 1D? For example, if Ku80 activity depends on CST-complex, as suggested in Figure 1C, why does Ku80 knockout have a much more dramatic impact on survival than Ctc1 knockout?

The reviewer is correct: ionizing radiation does not solely produce DSBs with 3' ssDNA protrusions, the physical properties of IR result in the formation of a variety of DSBs, ranging from a blunt or near-blunt composition to DSBs with 5' or 3' protruding ssDNA segments of different lengths (Sage & Harrison, 2011)⁴. That was also one of the purposes to include the IR-survival data: to provide support for the notion that Ctc1 may be involved in the repair of only a subset of the IR-induced breaks that require Ku80-mediated NHEJ, leading up to demonstrating that CTC1 is required for repair of Cas9-induced breaks with 3' protrusions, but not of breaks with 5' protrusions. This in contrast to Ku, which is needed for NHEJ repair of the entire spectrum of DSBs.

It has not been our intention to correlate the reduced mutation frequency observed in (old) Figure 1C to the sensitivity of the same cell-lines in (old) Figure 1D and we apologize if that is how it came across. We have now moved Figure 1C to Figure 2E, where we believe it is more logically placed and discussed in the text, which we have amended accordingly (lines 127 – 134)*.

4) The Authors demonstrate that tandem duplications depend on Pol alpha for DSBs that give rise to protruding 3' ends but not in the case of protruding 5' ends. This is an interesting finding, that should be further explored. Also, in the case of the 5' protruding ends, the double knock-out (*Polq*^{-/-} *CTC1*^{-/-}) increase TDs, while this is not the case of the single knock-out. How the Authors would explain this finding? Finally, why in these conditions Ku80 activity is independent on CTC?

We agree that one of the interesting findings of our study is that the involvement of Shld-CST-Pol α is specific for DSBs with 3' protruding ends, a feature that we now further validated in human cells, i.e. RPE1-*Rev7* knockouts (Fig 3h-3i and Fig S6h-j).

As for why the double knockouts appear to have increased TDs, we wish to copy the text we have written above in response to reviewer #1:

“We suspect that also here the way the data are presented may lead to some confusion. It is indeed true that in the *Polq*^{-/-} *Ctc1*^{-/-} double knockout cells, the relative contribution of TDs

increases. However, the absolute frequency of mutation in the double knockouts is ~ 35% of that of wild-type cells. It may thus be argued that the absolute frequency of TDs is not affected (~30% of ~0.35 MF versus ~10% of ~1.00 MF).

This reasoning nevertheless defers the question to something that is also unclear to us: the observation of a lower mutation frequency in *Polq*^{-/-} *Ctc1*^{-/-} double knockout cells compared to *Polq*^{-/-} cells, which argues that Ctc1 also has some suppressor activity on mutation induction in *Polq*^{-/-} cells at 5' overhangs. At present, we can only speculate: one, to us reasonable, explanation may be that for a subfraction of DSBs (in yet to be defined cellular contexts, e.g. cell cycle phase), the 5' protruding end is subject to (excessive) end-resection, which can be countered by Ctc1-mediated fill-in of the newly arisen 3' overhang followed by NHEJ. Because the end-resection removes the overhangs, this outcome will not manifest as a TD but as a deletion in a mutation profile. In the absence of TMEJ (*Polq*^{-/-}) and of Ctc1-mediated fill-in/NHEJ it may be that specifically these DSBs are unreparable (CST-mediated fill-in is necessary for 3' overhang repair by NHEJ and TMEJ is necessary for 3' overhang repair by TMEJ) and cells containing these DSBs either die or at least fail to proliferate, hence dropping out of our analysis. This would explain both a decreased mutation frequency as well as a relatively more affected drop in deletions (giving the impression of more TDs) for double mutant cells. This possibility has now been mentioned in the result-section as a potential explanation for the observed spectrum in those cells (lines 118-122)*."

As for why TDs occurring at breaks with 5' overhangs do not require CST, we envisage de novo DNA synthesis at the 3' terminus templated by the 5' protrusion (which can be achieved by multiple eukaryotic polymerases) producing a blunt template that can be used by NHEJ - the "fill-in" on this substrate can occur without the need for a new primer to be generated.

5) The Authors conclude "our study demonstrates that besides the recently described role in preventing the loss of genetic information around DSBs, the 53BP1-Shieldin-CST axis contributes to genome expansion by Pol α -Primase mediated fill-in synthesis of complementary 3' ssDNA protrusions resulting in small tandem duplications" (line 172). However, the Authors have not directly shown the activity of Pol alpha nor have evaluated the extent of the 3' protruding end which they hypothesize to be shorter, thanks to the activity of the Primase. To demonstrate that tandem duplications arise from Pol alpha activity the Authors should demonstrate that ssDNA extension is indeed mediated by Pol alpha. As a matter of fact, Pol alpha may act as a scaffold, without a direct engagement.

We were confused by this text, especially the usage of the wording "the authors should demonstrate that ssDNA extension is indeed mediated by Pol-alpha". This phrasing seems to suggest that we propose that Pol-alpha extends the 3' protruding-ends that are generated by Cas9-N863A induced breaks. This is, however, not the case. Our data (and model, Fig 4j) shows that Pol-alpha initiates near the end of the 3' overhang, to subsequently perform fill-in synthesis using the existing 3' overhang as a template, in a 5' to 3' direction. Because of this activity, the protruding ssDNA ends become double-stranded and a substrate for NHEJ.

6) The authors fail to demonstrate a direct involvement of Pol α -Primase in tandem duplication. They claim that "Providing direct evidence is challenging as Pol α is essential for genome duplication" (line 144). The role of Pol α -Primase in tandem duplication could be tested by arresting cells in G2 before addition of Pol α -Primase inhibitors.

We thank the reviewer for this suggestion. In previous attempts, prior to our first submission, we had difficulty with obtaining experimental genetic conditions that allowed for Pol-alpha inhibition. Motivated by the comments of the reviewers we tried several approaches combining cell-cycle synchronization, Cas9-break induction, and Pol-alpha inhibition. We have now included experiments using a light-inducible³ sgRNA to instantly make DSBs in synchronized cells. By including the Pol-alpha inhibitor CD437 in this experiment, we now show a strong reduction in the amount of TDs upon inhibition of this polymerase (Figure 4A-B).

Minor revisions:

1) The Authors provide references to support the role for tandem duplications that are either outdated, or not pertinent (in particular references 1-5). A more detailed, and comprehensive description and appraisal of the role of TDs in human disease is warranted, updated references.

We are not sure which references/literature the reviewer refers to: to our knowledge, we have included the most relevant literature on the formation of **small** DNA duplications; there is simply not much known concerning their etiology (we have now included a few extra references where ITDs in oncogenes are reported⁵⁻⁷). Perhaps, the reviewer refers to longer TDs or TDs in general, hence, we have now included a sentence (including references) in the introduction to discriminate between small duplications (e.g. ITDs) and larger chromosomal segment duplications (1-100 kb), which are associated with the 'BRCAness' signature⁸⁻¹⁰ (lines 33-35)*.

2) The introduction is too succinct and fails to provide comprehensive information on the TDs and the mechanisms eliciting their appearance.

We hope that our comment at the previous point also covers this point.

3) Why in figure 1E and 1F data the data for the double knock-out (Polq^{-/-} CTC1^{-/-}) are not included?

This is because the mutation frequency in those cells is nearly abolished (see figure 1d). Sorting of mutant (GFP-negative) cells in this background does not give enough cells to extract DNA for NGS using our standard workflow. We have performed NGS on unselected samples to also pick up potential mutations in this background, but that provided only few mutant reads, which had characteristics that are reminiscent of PCR- and NGS-related noise that we've observed in e.g. non-targeted controls .

4) The sentence on line 65-66 "indicating that CST is epistatic to Ku80 in 66 processing DSBs with 3' overhangs (Figure 1C)" is unclear.

Please see our response at 'specific point 2' on this matter.

5) The sentence on line 70 "The latter suggests that the CST-complex is involved in processing a specific subset of NHEJ substrates" is not supported by the data shown.

We agree with the reviewer that at this position in the text, this sentence was not supported by the data shown up till that point. Therefore, we have moved this sentence towards the end of the second results paragraph ('Contribution of CST to NHEJ is DNA break configuration specific') where we think it is reasonable to put this suggestion forward because of the different involvement of the CST-complex in Cas9-N863A and Cas9-D10A-induced breaks, and the milder sensitivity of CST-

deficient cells to ionizing radiation as compared to Ku80 deficient cells.

6) To help the reader understand the experiments performed, more details should be provided between the two loci, HPRT-eGFP and HPRT exon 3.

We have now provided more information at the start of the first results section and we have also updated figure 1 (1A and 1B) and figure S1 (S1B) to provide more detail on the generation of the HPRT-eGFP tagged gene.

7) Based on comparison between HPRT-eGFP and HPRT exon 3, it seems that DSBs resolution and CST contribution depends on sequences surrounding DSBs. How frequent in the genome and therefore relevant is the situation represented by HPRT-eGFP?

We indeed observe a difference between the exon 3 and eGFP site when it comes to the number of TDs that remain in the absence of Ku80 or Ctc1. In wild-type cells, however, we argue that the contribution of the CST-complex to mutagenic-repair is quite similar for both sites, as indicated by the strong reduced mutation frequency and the clear shift in the mutational spectrum in both sequence-contexts. Although TDs make up for a larger fraction of the total mutant outcomes at the exon 3 site in the absence of Ku80 or Ctc1, their molecular parameters (*e.g.* the degree of microhomology present at the junctions (figure S3c) are different than for the majority of TDs found in wild-type cells, further supporting the conclusion that also at this site, Ku80 and CST-dependent TD-formation is the dominant mechanism. This phenomenon (likely) reflects how polymerase theta action, which can compensate for the loss of CST, is influenced by the sequence context surrounding the DSB.

We have now added a sentence to the end of the paragraph 'CST-complex promotes tandem duplication formation at breaks with complementary ends', to explain this better (lines 97-103)*.

With respect to the remark of *HPRT-eGFP* being a just representation, we did include additional target-sites to validate our findings (*i.e.* *FLT3* and *HBB*, Figure 3D-F and Figure S6C-E respectively).

8) On line 114 "Figure 3A shows a decreased mutation frequency upon loss of 53BP1 and Shld2, similar to that observed upon loss of Ctc1 and Stn1". References to figures are missing, in particular fig. 1 and S2C.

This has been changed

9) Line 119 "Similar results were obtained when targeting exon 3 of HPRT-eGFP: a strongly decreased mutation frequency and altered repair outcomes in Tp53bp1 and Shld2 knockout cells as compared to wild-type cells (Figure S5)." Please change the label Figure S5A with Figure S5A-E.

This has been changed

* Line numbers correspond to the manuscript with all changes accepted.

References

- 1 Bothmer, A. *et al.* Characterization of the interplay between DNA repair and CRISPR/Cas9-induced DNA lesions at an endogenous locus. *Nat Commun* **8**, 13905, doi:10.1038/ncomms13905 (2017).
- 2 Schimmel, J., Kool, H., van Schendel, R. & Tijsterman, M. Mutational signatures of non-homologous and polymerase theta-mediated end-joining in embryonic stem cells. *EMBO J* **36**, 3634-3649, doi:10.15252/embj.201796948 (2017).
- 3 Liu, Y. *et al.* Very fast CRISPR on demand. *Science* **368**, 1265-1269, doi:10.1126/science.aay8204 (2020).
- 4 Sage, E. & Harrison, L. Clustered DNA lesion repair in eukaryotes: relevance to mutagenesis and cell survival. *Mutat Res* **711**, 123-133, doi:10.1016/j.mrfmmm.2010.12.010 (2011).
- 5 Wegert, J. *et al.* Recurrent intragenic rearrangements of EGFR and BRAF in soft tissue tumors of infants. *Nat Commun* **9**, 2378, doi:10.1038/s41467-018-04650-6 (2018).
- 6 Yeh, Y. C. *et al.* AKT1 internal tandem duplications and point mutations are the genetic hallmarks of sclerosing pneumocytoma. *Mod Pathol* **33**, 391-403, doi:10.1038/s41379-019-0357-y (2020).
- 7 Barets, D. *et al.* Specific and Sensitive Diagnosis of BCOR-ITD in Various Cancers by Digital PCR. *Front Oncol* **11**, 645512, doi:10.3389/fonc.2021.645512 (2021).
- 8 Willis, N. A. *et al.* Mechanism of tandem duplication formation in BRCA1-mutant cells. *Nature* **551**, 590-595, doi:10.1038/nature24477 (2017).
- 9 Stok, C., Kok, Y. P., van den Tempel, N. & van Vugt, M. Shaping the BRCAness mutational landscape by alternative double-strand break repair, replication stress and mitotic aberrancies. *Nucleic Acids Res*, doi:10.1093/nar/gkab151 (2021).
- 10 Kamp, J. A., van Schendel, R., Dilweg, I. W. & Tijsterman, M. BRCA1-associated structural variations are a consequence of polymerase theta-mediated end-joining. *Nat Commun* **11**, 3615, doi:10.1038/s41467-020-17455-3 (2020).